# Designing a MOOC on Computational Thinking, Programming and Robotics for Early Childhood Educators and Primary School Teachers: A Pilot Test Evaluation

Lúcia Amante *, Elizabeth Batista Souza, António Quintas-Mendes and Maribel Miranda-Pinto

LE@D—Laboratório de Educação a Distância e E-Learning, Open University, 1000-013 Lisbon, Portugal; elizabeth.souza@uab.pt (E.B.S.); antonio.mendes@uab.pt (A.Q.-M.); maribel.miranda@uab.pt (M.M.-P.)
* Correspondence: lucia.amante@uab.pt

**Abstract:** This study focuses on developing and evaluating an online course aimed at preschool educators and primary school teachers. It presents a Massive Open Online Course (MOOC) on computational thinking, programming and robotics developed as part of the project "Laboratory for Technology and Programming and Robotics Learning in Primary and Preschool Education in Portugal (KML II)" The MOOC design was inspired by a blended learning model used in teacher professional development at the project's inception and incorporates theoretical-pedagogical models of MOOC design as well as theoretical models of online interaction in virtual educational environments. The course will be offered on the NAU platform, a Portuguese MOOC platform. A pilot test was conducted with a purposive sample that included both participants from the target audience of the course as well as national and international experts specialised in these domains. The evaluation included a Likert scale questionnaire survey and open-ended questions. The results aim to validate the MOOC's quality, including its structure, content relevance, proposed activities, and learning design. The findings provide evidence to improve the final version of the MOOC, contributing to its effectiveness and adequacy to the target audience.

**Keywords:** MOOC; learning design; programming; robotics; primary school; early childhood; pilot test evaluation





## 1. Introduction

Acquiring knowledge and skills in computational thinking, programming, and robotics is of the utmost importance for teachers in the current educational landscape [1]. These domains provide teachers with powerful tools to engage and empower their students by fostering critical thinking, problem-solving, and creativity. By incorporating computational thinking into their pedagogical practices, teachers can guide students toward a systematic approach to problems, encouraging logical reasoning and analytical competencies. Programming skills allow teachers to introduce students to coding and computing concepts, opening up a whole new world of creativity and innovation in the classroom [2]. In addition, understanding robotics allows teachers to integrate hands-on experiences and project-based learning, bringing abstract concepts to life and promoting cooperation and teamwork [3]. By embracing these themes, teachers are enabling 21st-century skills, preparing students for the demands of a digital society and equipping them with vital knowledge and tools to fit into an increasingly complex and technology-driven world [4].

In this sense, the curricular inclusion of topics such as computational thinking, programming, and robotics in the early years of schooling is becoming increasingly more pressing [5–8], requiring the learning of teachers and educators in order to achieve this integration [9–11]. In the context of the Project "Laboratory for Technology and Programming and Robotics Learning in Preschool and Primary Education in Portugal" (KML II) [12], the

proposal of a Massive Open Online Course (MOOC) on computational thinking, programming and robotics was designed to expand professional learning and access to pedagogical strategies and resources in this domain for teachers in the Portuguese educational network and the entire Portuguese-speaking community. In fact, MOOCs have increasingly been established as an offer format suitable to the learning demands of this increasingly more digital global society that calls for open educational practices capable of reaching a wide range of people [13,14].

The design of the MOOC, which is presented in this article, is the result of ongoing professional development that took place in the blended learning format as part of the KML II project. The evaluation of this programme [15] made it possible to identify the learning profile of the potential target audience of the MOOC to be developed, giving support to its design process [16]. However, we would like to point out that rapid transformations, especially those arising from the COVID-19 pandemic, specifically the rapidly changing skills profile of the target audience, had an impact on the initial premises for developing this course. Initially, the research team had concerns about participants' digital skills, based on a survey conducted as part of the KML II Project [10], which indicated poor digital literacy and a preference for face-to-face or semi-face-to-face courses. However, the pandemic has caused some processes to speed up, thus requiring this audience to develop quickly, as well as pick up a variety of technological skills and a greater acceptance and familiarity with digital learning environments. We believe that this aspect will have a positive influence on the uptake of this type of professional development program and on the achievement of its objectives.

This text presents the results of the pilot test evaluation of the MOOC on Programming and Robotics in Preschool and Primary Education, developed by a group of expert researchers who were part of the KML II project and later validated by other researchers at partner institutions. For this pilot test, we considered an intentional sample composed of two participant profiles: content experts, teachers, and researchers in the area of computer science and educational technology, and individuals whose profile corresponds to that of the potential target audience, i.e., early childhood educators and primary school teachers.

The test was intended to assess the design and the contents developed for the MOOC, with the purpose of collecting contributions that would allow adjustments and improvements to the final product being offered. In the first part of this article, we present the development context of the KML II project and its objectives, followed by a section on the development of the learning design of the MOOC, the decisions taken, and the reasoning behind them. Next, we briefly present the methodological options adopted in the process of testing and processing the data reporting on the type of questionnaire constructed for this purpose and the data analysis methods adopted. We then present the quantitative and qualitative results collected in this validation. Finally, we interpret, discuss, and take stock of the results, reflecting on them and their implications. In the conclusion, we put forward possible developments for the design of the MOOC to be made available shortly on the NAU platform, which is a Portuguese platform that was created by the Fundação da Ciência e a Tecnologia (FCT) and which is configured as a platform specially created to support education and professional development for large audiences.

### 1.1. Context of the KML II Project

The project "Laboratory for Technology and Programming and Robotics Learning in Preschool and Primary Education in Portugal" (KML II) was developed between 2018 and 2022 [12]. It is important to mention that this project is a continuation of the Kids Media Lab project (https://www.nonio.uminho.pt/kidsmedialab, accessed on 12 July 2023), an individual post-doctoral project, which was developed in five districts in the north of Portugal and focused solely on preschool education, in order to understand how children learn to program at this age. The need to accompany children in the transition phase to primary education raised the possibility of an extension of the project to another

educational level but also of its expansion at the national level due to interest in this area in educational curricula.

In this sense, the KML II project has attempted to study how to bring computational thinking, programming, and robotics to preschool and primary education in order to produce a specific proposal for integration in these educational settings across all areas of knowledge. Based on the research carried out, we have developed a b-learning professional development programme in the area that is the basis for the design and construction of the MOOC, and which is also based on a study extended across Portugal about the professional learning needs of Early Childhood Educators and Primary Education Teachers [10].

Providing learning in this area to teachers and future teachers is crucial for the development of interchangeable skills that fit all the areas of knowledge and contents of preschool and primary education, as explained in the Curricular Guidelines and Essential Learning made available by the Directorate General of Education in Portugal [17]. Furthermore, also promoting the participation of children in the activities foreseen in this project was an underlying principle of our theoretical approach that, together with others, guided the design of the MOOC [16], which will be available to all education professionals on the NAU open platform. The implementation of the KML II project enabled the development of creative programming activities and educational games, both using and not using technology, integrated into the curricular areas. These initiatives promote the active participation of children and encourage creativity, critical thinking, problem-solving, and cooperative work. We noticed through the involvement of children and the facilitating environments that promote these activities that it is possible to achieve the 5 "C's" recommended by Marina Bers in the "Positive Technological Development" model [18] (p. 6), in which we intend to promote environments that provide children with "Competence; Connection; Character; Confidence; Caring; Contribution", through the integration of technology and programming from a healthy and positive perspective, which will influence their development.

*1.2. Design of the KML II MOOC*

In this section, we will provide a relatively detailed description of the objectives, contents, modules, and pedagogical foundations of the course that has been proposed. This detailed description is intended to correspond to the need felt by several authors, such as Patton [19] and Fawnes and Sinclair [20], to resort to a "thick description" of the processes of characterisation and evaluation of courses so that these processes are not limited to superficial indicators about them and allow a deeper understanding of the pedagogical assumptions underlying a course, as well as the institutional, disciplinary, and social constraints that may affect it [16].

As a starting point for defining the design of this MOOC, the objective described in the scope of the KML II Project was taken into consideration. Thus, as a final outcome of the project, an activity was designed in which a MOOC prototype was developed to support professional development in Portugal as a means of expanding the b-learning program previously developed with a limited group of participants. Currently, this network includes 17,064 early childhood educators and 30,986 primary school teachers, according to 2021 data from Pordata.pt (https://www.pordata.pt/db/portugal/ambiente+de+consult/table, accessed on 12 July 2023). The Portuguese education system includes twelve years of compulsory schooling. Nine years of basic education (divided into first, second, and third cycles) and three years of secondary education. Before compulsory schooling, preschool education is offered from three and five years of age, after which primary education begins, lasting four years (first cycle).

Taking into account this information, associated with the results of the aforementioned b-learning experience and the conclusions of the study and analysis of teacher professional learning needs in Portugal [10], it was understood that the MOOC should, in its format, combine elements of content-based transmission, task-based performance, and network-based creation, with the purpose of achieving the proposed objectives. It should

be highlighted that part of this design was tested in preschool and primary education settings during the data collection stage of the project [17].

In this sense, the design that has been proposed for the KML II MOOC is based, in pedagogical terms, not only on the principles of a cMOOC (student-centred and featuring peer collaboration) nor on those of an xMOOC (centred on interaction with contents and self-learning), contemplating typical elements of each of these two types. Thus, the proposed MOOC, despite consisting of materials designed and prepared in advance, such as videos, texts, and activities, typical of xMOOCs, seeks to introduce some connectivist participation practices, adhering to Downes' four principles of [21] autonomy, connectedness, diversity, and openness. These last principles are guaranteed by the types of activities proposed, which promote interaction and sharing among participants, as well as by the licenses applied to the resources provided [16] (p. 257).

Regarding the interaction strategies of the course, Anderson's Interaction Equivalency Theorem (IET) [22] was used as a reference to determine the most appropriate type of strategy. Based on the types of interaction proposed by Moore [23]—student–teacher, student–content, and student–student—and the types added by Anderson and Garrison [24]—teacher–teacher, teacher–content, and content–content—an analysis of the possibilities of interaction was carried out. The IET was applied with a view to identifying the priority form of interaction in the course design, seeking to guarantee high levels of meaningful and deep learning. In this, we followed Miyazoe and Anderson [25] when they pointed out that the IET was created with the purpose of providing "a theoretical basis for judging the appropriate amounts of each of the various forms of possible interaction" (p. 2).

In the KML II MOOC, we opted to prioritise student-content interaction, offering materials and activities that promote self-learning without the presence of teachers or tutors for guidance. Student-student interaction is also stimulated through shared spaces and encouragement of interaction among colleagues. Thus, the IET was a tool used by the development team to assist in making decisions related to the course design [22].

It is also important to note that the design of the KML II MOOC was faced with several constraints, including the availability of time and human resources, and limitations partly justified by the disruption of the project schedule due to the COVID-19 pandemic [16]. The MOOC entitled "Programming and Robotics in Primary and -Education" was organised into five modules, with an estimated dedication time load of 25 h, as shown in Figure 1.

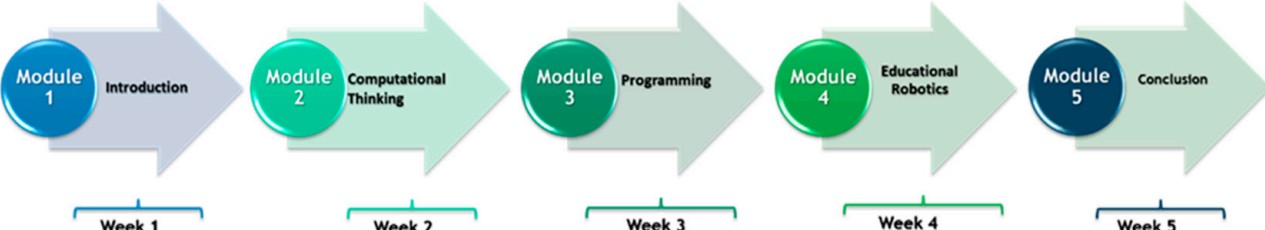

**Figure 1.** Basic Structure of the MOOC.

The purpose of the MOOC is to promote the development of essential competencies so that childhood educators and primary school teachers can integrate computational thinking, programming, and educational robotics strategies into their teaching activities. In order to achieve this goal, three competencies were defined, which refer to identifying opportunities and planning activities that integrate computational thinking, programming, and educational robotics.

Therefore, the MOOC of the KML II project submitted to the pilot test is structured as shown in Table 1.

**Table 1.** Structure of the MOOC of the KML II Project.

| Weeks | Module 1—Introduction | | | |
|---|---|---|---|---|
| | **Objectives** | **Topics** | **Resources** | **Activities/Assessment** |
| Week 1 | Introduce participants to the NAU platform and to working on a MOOC. | Learning with a MOOC | • Welcome video and text explaining how to learn with a MOOC. | • Forum Presentation<br>• Quiz |
| | Present the formative path of the course, methodology, assessment process, timetable, etc. | The course | • Course plan<br>• Timetable | • Self-diagnostic inventory<br>• My e-portfolio (part 1)<br>• Community forum |
| Week 2 | **Module 2—Computational Thinking (CT)** | | | |
| | **Objectives** | **Topics** | **Resources** | **Activities/Assessment** |
| | Identify opportunities to use computational thinking (CT) in activities provided in the school curriculum. | CT | • Video and text with the "Principles of CT with/without the use of technology" | • CT Forum<br>• Tests with automatic answer |
| | | Planning CT activities | • Base texts and supplementary Reading<br>• Artifacts with CT activities | • CT activities without the use of digital technology:<br>  ✓ Sequences<br>  ✓ Concept map<br>  ✓ Home-School route<br>  ✓ Cooking recipe<br>• CT activities with the use of digital technology:<br>  ✓ The hour of code<br>• Planning a CT activity:<br>• My e-portfolio (part 2) |
| Week 3 | **Module 3—Programming** | | | |
| | **Objectives** | **Topics** | **Resources** | **Activities/Assessment** |
| | Identify opportunities to use programming in activities planned in the school curriculum. | Principles of Programming | • Video and text "Principles of Programming" | • Programming Forum<br>• Automatic response tests |
| | Develop a programming project aimed at preschool learning. (A proposed pathway for early childhood educators) | Programming language applied to preschool (ScratchJr) | • Basic text about ScratchJr programming<br>• Planning of activities | • Identifying ScratchJr Algorithms<br>• ScratchJr Programming Cards<br>• My first ScratchJr Project |
| | Develop a programming project aimed at learning in primary school (Proposed pathways for primary teachers) | Programming language applied to primary education (Scratch) | • Basic text and videos "Programming with Scratch" | • Scratch community registration<br>• "Learn to program or program to learn"<br>• "Let's Dance" Project<br>• Labyrinth Game Project<br>• Finding the error project<br>• Building original Scratch project |
| | Learn about other programming languages that apply to preschool and primary education | Other programming languages | • Texts for further reading | • Research of other languages available by participants for discussion in forum<br>• Quiz<br>• My e-portfolio (part 3) |

**Table 1.** *Cont.*

| Weeks | Module 1—Introduction | | | |
| --- | --- | --- | --- | --- |
| | Objectives | Topics | Resources | Activities/Assessment |
| Week 4 | Module 4—Educational Robotics | | | |
| | Objectives | Topics | Resources | Activities/Assessment |
| | Identify opportunities for using robotics in the activities planned in the school curriculum. | Robotics in education | • Videos and text about robotics in education<br>• Types of robots and how to use them in an educational context | • Educational robotics forum<br>• Quiz |
| | Set out educational activities using robotics to develop skills set out in the curriculum. | Planning educational activities with the use of robots | • Apps and games for offline or online robotics activities | • Coding game off (*CodyRoby*)<br>• Online robot (*Bee-bot*)<br>• Planning of educative robotics activity<br>• My e-portfolio (part 4) |
| Week 5 | Module 5—Conclusion of the MOOC | | | |
| | Objectives | Topics | Resources | Activities/Assessment |
| | Identify opportunities to use computational thinking, programming, and robotics in activities provided in the school curriculum. | Retrospective | • Course overview text<br>• Videos with testimonials from educators and teachers who participated in the KML II Project. | • Quiz<br>• Evaluation of my e-portfolio (part 5) |
| | Monitor participant satisfaction and collect data to further improve the educational programme. | Evaluating educational programme and student participation | • Google Docs forms | • Self-assessment test<br>• Final questionnaire |

The design proposed here also took into consideration the 10 principles established by Guàrdia, Maina, and Sangrá [26] for the development of a MOOC, as shown in Table 2.

**Table 2.** Principles of Guàrdia et al. [26] applied to the KML II MOOC.

| Principle | Elements MOOC KML II |
| --- | --- |
| Competency-based design | The MOOC was planned based on an objective and three competencies that are fully in line with the daily lives of teachers and educators in their work environment. All the resources, including videos and texts, provide real life case studies and the planned activities guarantee authentic experiences to those carried out in an educational setting. |
| Empowerment of learners | The design puts participants at the centre of the learning process, allowing each one to establish the path they want to follow, with space to define their own personal goals. An example of this is the fact that in Module 3 (Programming) the MOOC offers two possible pathways, one for early childhood educators and another for teachers.<br>The adoption of the e-portfolio allows for self-regulation of learning.<br>Themed forums are adopted as privileged spaces to provide and obtain peer support. |
| Clear learning plan and guidelines | In addition to the course plan, presented at the beginning, the MOOC includes a series of guidelines at each stage of the course, with the first week having the function of introducing participants to, the MOOC, setting expectations, and providing indications about the planned learning paths, both for childhood educators and for teachers. |
| Collaborative learning | Even though the design is quite flexible and allows for a fairly autonomous trajectory, strategies enabling interaction among participants were valued, such as the forums created for each theme and explicit guidance on efficient ways of communicating and interacting with peers. In addition, the creation of a space for sharing and interaction between participants (collaborative e-portfolio MOOC KML II) was also emphasised. |

**Table 2.** *Cont.*

| Principle | Elements MOOC KML II |
| --- | --- |
| Social networking | The creation of learning communities is especially encouraged in the activity of creating an e-portfolio, as it pervades the whole journey through the MOOC and the sharing of productions and appreciation of the work of peers is encouraged, leaving them comments. |
| Peer cooperation | Peer cooperation is encouraged, either in the questions left in the forums to stimulate debate or, more explicitly, in the peer assessment activity when they are asked to analyse the work of some colleagues, leaving them comments and contributions. |
| Quality criteria for knowledge creation and generation | The MOOC design encourages the search for content external to that provided on the platform and the production and sharing in Web 2.0 applications. To this end, guidelines on good practices in virtual environments and care in the use and sharing of content were included regarding security and rules for using content available on the web. |
| Interest Groups | In the design, there is a space specially designed for the formation of these groups, in a forum called "Community Forum", in which there are no predefined themes and where interaction is encouraged whenever participants wish to initiate a discussion around a theme that does not fit specifically into the themed forums that have already been created. |
| Peer assessment and feedback | The assessment process is based on self-assessment activities with explanatory automatic responses (tests and quizzes), self-assessment (quizzes and e-portfolio), and peer assessment (e-portfolio). |
| Enriched learning with media technology | The themes around which the MOOC was organised promote the use of many different types of technology and applications for carrying out activities. Moreover, the organisation of the e-portfolio, in which the use of a Web 2.0 application is encouraged, offers participants a variety of means of communication beyond the platform spaces. |

It should also be noted that the development work of this course took into consideration the steps indicated in the Quality Reference Framework (QRF) for the Quality of MOOCs, prepared under the MOOQ Project of the European Union [27], which foresees a testing activity in the implementation phase with representatives of the target audience prior to the actual delivery.

## 2. Materials and Methods

Research on the testing process was conducted using a qualitative, non-experimental, and descriptive design [28]. Although we used an instrument with mixed characteristics, which included closed- and open-ended questions, the main focus was not on the number of respondents, nor on the generalisation of results, but rather on the possibility of obtaining data to describe and interpret the reality presented, in this case, the MOOC design, and on seeking to transform this reality, based on listening to the different voices in the test group. The aim was to validate and improve the design of the proposed course before opening it to the general public. This testing process is characterised as the last stage of MOOC development (Figure 2).

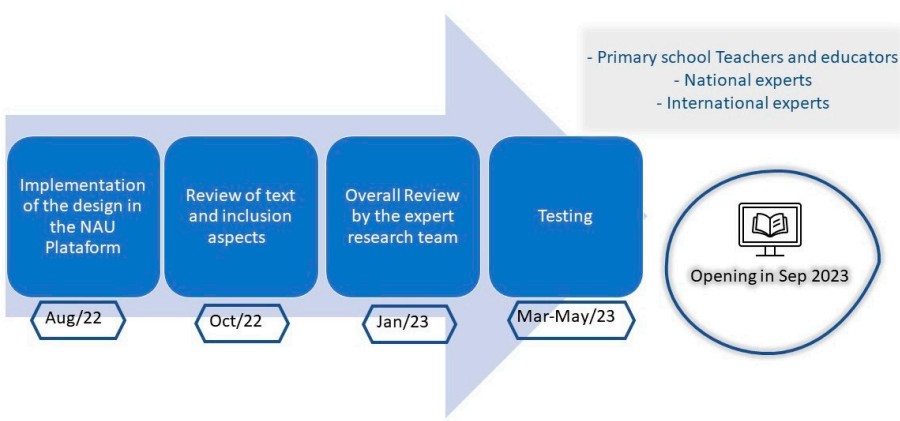

**Figure 2.** KML II MOOC development process.

For data collection, a questionnaire was used, which was answered based on a five-point Likert scale, where 1 indicates that participants "Strongly disagree" with the respective statement and 5 indicates that they "Strongly agree" with it.

The first part of the questionnaire included 12 items aimed at assessing general aspects of the MOOC. In the second part, the items were organised into blocks with a variable number of items relating to specific aspects of each of the course modules: Module 1, Getting Started; Module 2, Computational Thinking; Module 3, Programming; Module 4, Educational Robotics; and Module 5, Conclusions. Following each block of items, space was left for additional comments to allow participants to justify their answers and provide details on aspects to be improved, giving suggestions wherever possible.

A purposive sample was made up of seventeen participants: four childhood educators, four primary school teachers, and nine content experts, teachers, and researchers in the area of computing and educational technology, six from national higher education institutions, and three from Spain. Six participants were male, and eleven were female. The pilot test took place between March and May, with each participant being individually instructed to explore all the course modules, carrying out the activities in the sequence foreseen and encouraged to make critical comments and suggestions about them, according to the items detailed in the questionnaire.

For data analysis, descriptive statistical measures (frequency, mean, and standard deviation) were used for the closed-ended items of the questionnaire. For the open-ended responses, we used content analysis [29], the results of which were triangulated with the data from the closed-ended items of the questionnaire, allowing us to clarify and complement the perception expressed in scores resulting from the Likert scale. The purpose was to obtain an overview of participant perception of the MOOC proposal, as well as to identify weaknesses and suggestions for improvement.

## 3. Results

The results will be presented according to the questionnaire's structural logic, starting with the general aspects of the MOOC design and moving on to the specific aspects of the themed modules.

### 3.1. General Aspects of the MOOC

In the first part, general design features of the MOOC were evaluated, including navigability, format, resources, platform organisation, layout, time, complexity of the activities, etc. Table 3 summarises the analysis of the data obtained.

The data indicate a very positive opinion of the course, with mean scores generally above 4.5, considering the Likert scale in which 5 was the maximum value. The low spread of results should be highlighted, with standard deviation values always below 1, with the exception of the item regarding the length of the course, which registered a greater spread. The most valued features of the course are the easy navigation and the organisation of modules. Next, the coherence of the course contents and the organisation of the discussion forums stand out, as well as the writing of the text and the language used. The ease of use of the Padlet tool for creating the requested e-portfolio comes next, as well as the integration of the resources added to the course and the relevance of the "Learn more" shared content. Three items were rated lower, each below the average of 4.5: the appeal of the course design, the e-portfolio development activity aiming at sharing the production among the participants, and also the suitability of the time allotted to the course. The complexity of the activities also received an approval rating, which, on average, does not exceed 4.5.

**Table 3.** Data analysis of general features of the MOOC.

| General Features | *f* | $\bar{x}$ | $\sigma$ |
|---|---|---|---|
| The course was easy to navigate. | 17 | 4.8 | 0.53 |
| The way the course is designed is appealing. | 17 | 4.4 | 0.71 |
| The content seemed to be presented in a coherent way. | 17 | 4.7 | 0.59 |
| The additional resources (videos, texts, quizzes, hyperlinks) are well integrated into the course. | 17 | 4.6 | 0.49 |
| The creation of an e-portfolio is a relevant strategy for sharing the productions in this MOOC. | 17 | 4.4 | 0.80 |
| The Padlet tool for organising the e-portfolio is easy to use. | 17 | 4.6 | 0.61 |
| The division and sequence of the course modules seemed suitable. | 17 | 4.8 | 0.56 |
| The total time foreseen to carry out the MOOC seems suitable. | 17 | 4.4 | 1 |
| The complexity of the activities seems compatible with the profile of the target audience. | 17 | 4.5 | 0.51 |
| The text is well-written and uses language accessible to the target audience. | 17 | 4.7 | 0.47 |
| The organisation of the forums seems effective for holding the intended debates. | 17 | 4.7 | 0.47 |
| The contents shared in the "Learn more" units are relevant. | 17 | 4.6 | 0.79 |

The analysis of the open-ended responses confirmed and reinforced some of these appreciations, allowing the slight differences between the items presented in Table 3 to be clarified. The lower score regarding the appeal of the course design is perceived in some comments to be related to the "unprofessional" quality of some of the teachers' videos. The running time aspect was mentioned in the additional comment spaces by six participants as a factor to be taken into account, with some of them considering the course to be very demanding.

Six participants also questioned the relevance of the individual e-portfolio. Either because they considered the possibility of participating in the course with people with poor digital literacy, because they felt that not many participants would share and visit their colleagues' e-portfolios, or because they felt that it detracts from the central objective of the course. They suggest reviewing the strategy, with one participant proposing the creation of a "Community Padlet", as shown in the testimony of one of the experts:

> *Involving participants with poorer digital competencies and asking them to create and organise an e-portfolio may divert efforts from the main theme of the course, as there is already a space for sharing and reflection in the forums. Although the use of Padlet may lead to the fulfilment of learning, since the course is a MOOC, with predictably dozens or hundreds of participants, the idea is that these spaces (Padlet) will be of consultation and analysis almost exclusively of their creators, possibly it would be more interesting to create a community Padlet, with each person sharing their finished projects in an appropriate area (after discussion in the forums), so that at the end of the course they might have access to a diversified range of activities and reflections made during the course.* (LC Expert)

Regarding resources, the open-ended responses show that the lowest value assigned to this item by some of the participants is related to the lack of subtitles in Portuguese in some videos, particularly those referring to the Programming Module. There is also a feature mentioned by two participants that refer to the use of external platforms and the need for new records, which they consider may cause some confusion to the participants. In fact, in addition to Padlet, in the learning programme, participants are also required to download applications, access and register on websites, among other things.

There were also detailed contributions, specific suggestions on linguistic aspects, as well as the proposal to subtitle some icons used. In addition to these observations, criticisms, and suggestions, there were many comments praising the MOOC for its structure,

comprehensive content, accessibility, relevance, interactivity, and creativity, which confirms the positive assessment expressed by the score recorded in response to the closed items.

### 3.2. Evaluation of Module 1—Introduction

The second part of the form specifically addressed the elements that make up each of the five modules of the MOOC. As far as Module 1 (Introduction) is concerned, the results point to a very positive assessment (Table 4), with all items recording average scores of 4.7 or 4.8 out of 5. In the set of items, it is worth separating the first three, all of them related to the initial questionnaire, from the remaining four, related to information about the functioning of the course. Thus, we can see that the introductory module seems to fulfil its objectives well with regard to this information. The spread of results is minimal ($\sigma$ less than 1), with the highest spread being found in the last item, related to the assessment and certification of the course, indicating that some participants may not have been fully informed about this process.

**Table 4.** Data Analysis of Module 1—Introduction.

| Module 1—Introduction | $f$ | $\bar{x}$ | $\sigma$ |
|---|---|---|---|
| It was easy to complete the Initial Questionnaire in the estimated time. | 16 | 4.7 | 0.60 |
| The instructions given for completing the Questionnaire are clear. | 17 | 4.8 | 0.56 |
| I was able to submit the Initial Questionnaire without any problems. | 17 | 4.7 | 0.70 |
| The instructions for creating the e-portfolio are sufficiently clear. | 17 | 4.7 | 0.60 |
| The information provided in the introduction module is sufficient for understanding the course dynamics. | 17 | 4.8 | 0.58 |
| It was clear from the course information that there were two different routes recommended for teachers and educators. | 17 | 4.8 | 0.39 |
| The information about the evaluation process allows you to fully understand what needs to be done to obtain the certificate of participation. | 16 | 4.7 | 0.79 |

The analysis of the open-ended answers confirms the data in Table 4, allowing us to better elucidate some aspects. Thus, the information and instructions in the environment module were considered adequate, objective, and accessible. In this regard, the idea arose that not all of them should be completely visible, as they are exhaustive, and could, according to the suggestion, be consulted in full, if necessary. The completeness of the activity summary grid was also mentioned as something that could be improved/simplified to avoid demobilising participants. Regarding the e-portfolio activity, some participants suggested more specific instructions, as shown in the following statement:

> *I think that the portfolio activity should include more specific instructions on how it should be done. In particular, it would be important to add some questions to aid teacher reflection.* (Specialist GF)

### 3.3. Evaluation of Modules 2, 3, and 4

Table 5 summarises the results obtained in the blocks of items related to the themed course modules, given that these blocks consist of common items, although they are specific to each of the proposed activities. The values shown in Table 5 result from the average score attributed to the different activities proposed in each module. This data simplification was chosen because the scores of the different activities do not show significant differences. The particularities of the activities are better understood in the criticisms and suggestions left in the open questions for each themed module.

**Table 5.** Data Analysis of Modules 2, 3, and 4.

| Items /Modules | $f$ | $\overline{x}(M2)$ | $\overline{x}(M3)$ | $\overline{x}(M4)$ | $\overline{x}(M123)$ |
|---|---|---|---|---|---|
| The contents on CT seem relevant and well organised to me. | 17 | 4.8 | 4.8 | 4.9 | 4.8 |
| Activities on.... piqued my interest. | 17 | 4.7 | 4.8 | 4.9 | 4.8 |
| Instructions on the e-portfolio in Module 2 are clear enough. | 17 | 4.7 | 4.8 | 4.8 | 4.8 |
| Relevance of the activities for both routes. | 17 | 4.7 | 4.8 | 4.8 | 4.8 |
| Relevance of Educator Route activities. | 17 | - | 4.8 | - | 4.8 |
| Relevance of Teacher Route activities. | 17 | - | 4.9 | - | 4.9 |

Therefore, for modules M2 (Computational Thinking), M3 (Programming), and M4 (Educational Robotics), Table 5 shows that the average score obtained for each of the items is fairly consistent, with none of the modules standing out in any particular way. All of them scored very well, both in terms of content and of the activities proposed, both in those aimed at all participants and in those addressed to the route recommended to Educators or the route recommended to Teachers, included in Module 3.

In the analysis of the answers to the open-ended questions, the diversity of the contents covered, the relevance and adequacy of the activities, as well as the resources provided were emphasised in all three modules. Nevertheless, specific suggestions for improvement are made, in particular, on the indication of practices associated with Computational Thinking, as well as several suggestions about some activities so that they are better adapted to the different age groups of children, for example. Suggestions of support resources were also made, specifically about the elaboration of conceptual maps (required in one of the activities), and the adequacy of this activity to the objectives proposed in the activity is also questioned, while other possible strategies are suggested. Criticism is also raised regarding the inclusion of links to external platforms (complementary resources) that require payment, as well as to videos not subtitled in Portuguese, corroborating some aspects previously mentioned when assessing the general features of the MOOC. Other specific suggestions are also made, such as the connection between an activity in the Computational Thinking module (Conceptual Map) with a Programming activity, making the course more integrated, or suggestions to clarify the description of each of the activities, as exemplified below:

> *Perhaps it would be good to introduce some examples in the conceptualisation of computational thinking. It is quite dense, theoretical, and synthetic, and that could facilitate a better understanding. Activity 2. I am not sure that the concept map is the most appropriate tool for establishing the setting, characters, and narrative. Activity 5. I think I would include the learning objectives and assessment criteria in the template.* (SU Specialist)

> *The video in activity 4 enhances the activity by highlighting the importance of some designs for computational thinking. The examples would help to deepen the activities, as in the case with the planning activity.* (Specialist GF)

*3.4. Evaluation of Module 5—Conclusion*

We analysed Module 5 (Conclusions) separately, given its specific nature.

As shown in Table 6, the assessment of this module was also quite positive and consistent, with σ values below 1.

In addition to the positive assessment of the organisation of the contents, already covered in other items and which is assessed here accordingly, we highlight the good acceptance of the activity "Evaluate my e-portfolio", as well as the activity "Self-assessment".

**Table 6.** Data Analysis of Module 5—Conclusion.

| Module 5—Conclusions | $f$ | $\bar{x}$ | $\sigma$ |
|---|---|---|---|
| The contents of Module 5 are clear and well organised. | 17 | 4.8 | 0.56 |
| The "Evaluating my e-portfolio" activity is relevant. | 17 | 4.9 | 0.61 |
| The "Self-assessment" activity is relevant. | 17 | 4.8 | 0.39 |
| The final questionnaire seemed adequate; I had no difficulty in answering it. | 17 | 4.8 | 0.47 |

In the comments found in the answers to the open questions, the availability of a grading rubric for the evaluation of the e-portfolio was valued, considering that this instrument significantly clarifies the evaluation task to be developed. The most prominent criticism referred to the visual aspect of the course, which had also been mentioned in the evaluation of the general aspects of the course. Some participants consider that the videos made with the instructors could be more appealing. Nevertheless, most of the comments highly valued the MOOC as a whole:

> *Above all, I would highlight the organisation and structure of the course content. I think that it is very well organised, that the contents are very well sequenced, and that the materials are appropriate and allow the content to be learnt.* (Specialist GF)

> *Overall, this MOOC is excellently organised, and the topics are covered in a very motivating way. I think the resources added are very interesting, current, and appealing. Besides, they also provide a lot of scientific literature for those who want to know more.* (Cl.M. Educator)

**4. Discussion**

The objective of the MOOC testing was to validate the proposed learning design before the opening of its first edition. In this sense, the results of this process as well as its analysis and discussion by the team, will allow making alterations and adjustments that seem relevant for its improvement, namely by gathering contributions from both experts and subjects whose profile meets the characteristics of the target audience.

*4.1. On the General Aspects of the MOOC*

In global terms, the assessment was quite positive. The answers expressed in the closed items gave rise to high average values in most of the features under evaluation, particularly the navigation in the course and the organisation, nature, and sequence of the thematic modules. The open answers, in the format of free comments, confirmed this generalised positive perception registered in the participants' statements with adjectives such as relevant, interesting, well-managed, accessible, and complete. However, despite this generally very positive assessment, our concern was a finer analysis of the open answers that would allow us to uncover details and suggestions of the participants in the pilot test, pointing out issues to be taken into consideration by the MOOC development team, to help improve its design. Thus, in the general evaluation of the course, we identified a set of dimensions that we will now highlight and discuss.

The dimension execution time/complexity of the activities emerged as a factor to be taken into account, as the MOOC course was considered very demanding for the time set aside for it. Although this point is perceived as critical and deserves reflection by the team, we understand that this perception may have been affected by the fact that the participants had a short period to try out the MOOC and did not go through the process in the actual time allotted for the various activities.

The attractiveness of the course design also emerged with a lower average score than most of the items, having been the subject of comments in the open answers section. On the one hand, there are limitations arising from the basic layout of the platform that can hardly be overcome, but on the other hand, and we believe that this is the main focus

of criticism, some of the videos used are home recordings of the instructors, which do not exhibit a high degree of quality. We recognise this limitation, but we are faced with the impossibility of making professional recordings, either due to the availability of the trainers or the associated costs. We chose to use these recordings, given the relevance of their content, regardless of the lower quality of their form.

The dimension relevance of the individual e-portfolio was also questioned by some participants, and, similarly to the items related to the previous two aspects, it had one of the lowest mean scores of the set of assessed items. As previously mentioned, in the free comments, following the critique, a participant suggests the creation of a "Community Padlet". This suggestion deserves the best attention of the MOOC development team, considering valid arguments that the elaboration of individual e-portfolios may not be the best strategy to share the course output. The idea of a more collective strategy to make this sharing easier seems interesting to us.

As for the Resources dimension (videos, texts, questionnaires, hyperlinks), partly interconnected with the issue of the quality of the videos mentioned above, it also points to the issue of Portuguese subtitles in some of the suggested videos and the use of external platforms and the respective need for new records. We understand that this requirement may create some discomfort for participants with poorer digital skills. However, since we are working with a theme that requires the use of applications and other technological resources, even if we try to minimise this feature, we must assume that this course foresees the use of environments and resources external to the NAU platform and this is also a competence that should be developed in the target audience. We assume, in the design of the MOOC, the four principles of Downes [21], autonomy, diversity, openness, and connectedness, and part of them aims precisely to promote the creation of networks that go beyond the environment of the proposed course, promoting other interactions and sharing.

Indeed, increasingly, formative spaces are becoming more open and interconnected, expanding and complementing each other. In this context, it is crucial to have skills to navigate and establish connections between these spaces, to enhance their use according to the specific learning needs of each individual and each moment [30,31]. This dynamic reflects, to a certain extent, the relationship of the current society with technology. Therefore, it could be important to highlight this point in the introduction message of the course.

### 4.2. About the Different MOOC Modules

In the specific analysis of the evaluation of each of the modules, we verify that Module 1 (Introduction) seems to fulfil its objectives well, elucidating on the structure and organisation of the course, specifically on the two suggested routes (educator or teacher). We have noted that the two alternative routes proposed have been easily comprehended by all the participants.

The evaluation of the three themed modules has concentrated the highest number of contributions in the free comments, the three themes having been well evaluated for the quality of their content. As expected, a more critical and specific view was perceived by the experts. The teachers and educators were less detailed and critical in their considerations than the experts. We intended to obtain both perspectives: from the experts, who were predictably more incisive, but also the contribution of potential members of the target audience who could alert us, among other aspects, to the clarity and pertinence of the topics discussed and the activities proposed, according to their practice in the field with children.

In the last module, Conclusion, we also registered a positive evaluation, where it is pertinent to highlight the good acceptance of the activity "Evaluate my e-portfolio", as well as the activity "Self-evaluation". It should also be noted that comments valued the availability of a rubric grading for the evaluation of the e-portfolio. This is important in the evaluation context of any course but gains even more relevance in the context of courses such as MOOCs, in which the mechanisms of self-regulation of learning assume particular relevance to enhance the autonomous learning path [32,33].

*4.3. Summary*

In summary, we emphasise that, in addition to a generally very favourable assessment, several contributions were left that helped the development team to improve the proposal. A good part of these contributions will result in adjustments to the course, which we consider very relevant for its final version. Particularly noteworthy is a structural adjustment to the initially proposed design, which will lead to the creation of a community-sharing environment, the "Community Padlet", probably replacing the development of individual e-portfolios.

In other words, the strategy of creating personal learning environments will lead to the creation of a community for sharing plans, products, and projects carried out during the course, considering that this strategy seems to be more adjusted to the expectations of the target audience.

This alteration has an impact on the assessment proposal, which in the original design was based on a self-assessment process using a grading rubric, which included aspects related to the personal trajectory in the course. In the new perspective, the focus should be not only on self-evaluation but on the evaluation of the collective construction of the group, which will require an adaptation of the rubric, which will now include items that refer to interaction and collaboration with other participants. These changes are now being considered by the team, which intends to maintain the focus on continuous self-regulation, which will be joined by peer regulation resulting from sharing within the group in the collective space to be created. We believe that this can also expand and stimulate learning of a collaborative nature that can be advantageous for different profiles of participants [34].

**5. Conclusions**

According to Donald et al. [35], the concept of learning design encompasses both a procedural aspect and a tangible output. Learning design as a process encapsulates the activities undertaken by educators and learning designers. On the other hand, learning design as a product refers to the result that emerges from these activities. The authors encapsulate this twofold interpretation by defining learning design as

> *"A product (that) documents and describes a learning activity in such a way that other teachers can understand it and use it (in some way) in their own context. Typically, a learning design includes descriptions of learning tasks, resources and supports provided by the teacher. Learning design is also the process by which teachers design for learning, when they devise a plan, design or structure for a learning activity"* [35] (p. 180).

In the present paper, we widely document, through a relatively detailed description and pilot evaluation, the objectives, contents, modules, and pedagogical foundations of a MOOC on computational thinking, programming, and robotics. This detailed description and evaluation are intended to respond to the need felt by several authors, such as Patton [19] and Fownes and Sinclair [20], to use thick descriptions of the processes of characterising and evaluating courses so that these processes are not limited to superficial indicators about them, but allow a deeper understanding of the pedagogical assumptions underlying a course and its development, its structure, contents, materials and activities, as well as the the institutional, disciplinary and social constraints that may affect it.

Teachers and learning designers commonly use an iterative approach to building courses and learning resources. The present paper documents a specific phase in the development of a product (a MOOC) that we hope can be useful for other teachers, designers and teams involved in the conception, development, and implementation of MOOCs. It is the follow-up of other works in which we have described and evaluated an initial blended course that has inspired the present MOOC [15] and another work [16] in which the design of the MOOC was analysed in terms of its socio-technical context, describing the formal and informal interactions between the different actors involved in the design process in attempting to define a common vision and consensus, as well as the divergences and contradictions that are part of the learning design process. This continuous process of deeply documenting

the development and evaluation of a course corresponds to the emphasis on processes of description in social sciences in general [36] and in education in particular [20]. Specifically, in what relates to the development of MOOCs, we can find similar approaches of detailed description in the works of Donald et al. [37] and in the works of Freire [38,39]. These processes can also be linked with the concept of developmental evaluation [19], which involves the establishment of relationships and interactions between evaluators and project or program staff. In the context of developmental evaluation, the mobilisation of evaluators encourages the continuous collection, analysis, and use of data by project and programme managers and staff, with the aim of reinforcing ongoing decision-making processes. Given the continuous process of evaluation, a continuous flow of feedback becomes feasible, consequently allowing for ongoing adjustments to be made to projects and programs. Thus, developmental evaluation emerges as an appropriate method for initiatives conducted in complex or uncertain situations where the necessity for evidence-based decision-making persists throughout the project or program's life cycle.

In conclusion, the development of this Massive Open Online Course (MOOC) has been a dynamic and intricate process that has provided valuable insights into the realm of online education, and that allowed us to gain a comprehensive understanding of the multifaceted nature of this educational initiative. All the process was instrumental in guiding the iterative and adaptive nature of our MOOC development. This approach has enabled us to navigate the complex and evolving landscape of online learning, allowing for adjustments and refinements that respond effectively to the needs of both learners and instructors. The principles of ongoing feedback and continuous improvement have been seamlessly integrated into our development process, fostering a learning environment that is not only innovative but also responsive to the nuances of modern education. The pilot test presented here has clearly enhanced our ability to refine the MOOC. As we conclude this paper, we recognise that the development of a MOOC is not a linear process but a living entity shaped by diverse influences. By adopting a strategy of continuous evaluation and development, we go beyond static evaluations to engage in a continuous dialogue with the various players called upon to intervene in the process of developing a massive online course. This not only advanced our understanding of online course development but also opened up avenues for further research, exploration, and innovation in the dynamic field of online education, particularly with regard to the documentation processes so necessary for other researchers, designers and teachers who may be interested in getting involved in similar massive online course development processes.

**Author Contributions:** Conceptualisation, L.A., E.B.S., A.Q.-M. and M.M.-P.; Methodology, E.B.S. and M.M.-P.; Validation, L.A., E.B.S. and M.M.-P.; Formal analysis, L.A. and E.B.S.; Investigation, L.A., E.B.S., A.Q.-M. and M.M.-P.; Data curation, L.A. and E.B.S.; Writing—original draft, L.A., E.B.S., A.Q.-M. and M.M.-P.; Writing—review and editing, L.A., E.B.S., A.Q.-M. and M.M.-P.; Supervision, L.A.; Project administration, M.M.-P.; Funding acquisition, M.M.-P. All authors have read and agreed to the published version of the manuscript.

**Funding:** This research was funded under the project KML II—Laboratory of Technologies and Learning of Programming and Robotics for preschool and primary school, which is co-funded by FEDER through the COMPETE 2020—Operational Thematic Program for Competitiveness and Internationalization (POCI) and national funds through FCT—Portuguese Foundation for Science and Technology under project reference number PTDC/CED-EDG/28710/2017.

**Institutional Review Board Statement:** The study was conducted in accordance with the guidelines of the Ethics Committee for Research in Social and Human Sciences (CEICSH) of the University of Minho and the Ethics Committee of the Distance Education and eLearning Laboratory (LE@D) of the Open University.

**Informed Consent Statement:** Informed consent was obtained from all the study participants.

**Data Availability Statement:** The data presented in this study are available upon request to the corresponding author. The form used as a checklist for participants who tested the course is avail-

able at https://docs.google.com/document/d/1HNzD84CJ0cRqMz4kbTuuf0Ssi1wJDTeZ/edit?usp=sharing&ouid=103809363044589129049&rtpof=true&sd=true (accessed on 12 July 2023).

**Acknowledgments:** The authors would like to thank the educators, professors, and experts that collaborated on the Pilot Test. In addition, we acknowledge the work of partners and supporters (information on the project website https://www.nonio.uminho.pt/kml2/parcerias/, accessed on 12 July 2023), as well as in-kind donations from Clementoni, who provided robot kits. Further, we acknowledge the Distance Education and eLearning Laboratory—LE@D-UAb.

**Conflicts of Interest:** The authors declare no conflict of interest.

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
