# Peer review of "Designing a MOOC on Computational Thinking, Programming and Robotics for Early Childhood Educators and Primary School Teachers: A Pilot Test Evaluation"

_education, doi:10.3390/educsci13090863_

Round 1

Reviewer 1 Report

Thank you for letting me read this interesting paper. It was well-structured and easy to follow. My main concerns are below: 

1) The first paragraph argues for the value of computational thinking/programming and robotics base on only one reference - more are needed.

2) The use of the word 'nursery' needs to be defined in terms of age of children as it is different in many countries. I would also recommend to try and use the word 'preschool' (as in the title of the project) or early childhood education. Also, some information on the educational system (EC and primary) in Portugal would be useful for helping the reader understand the context.

3) Better to avoid the use of the word 'training' for teachers and replace it with professional development or professional learning to be consistent with relevant literature. 

4) The authors mention a range of scholars and their work on designing a Mooc (e.g., Anderson, Moore on page 4 - and then Guardia et al. on page 6). I would suggest deleting the ones  that are not used in the presentation of the paper. Only Guardia et al. are sufficiently explained and related to the MOOC. Please also check the use of IET - TEI on page 4.

5) There is no demographic info about the participants - only their gender. More details are necessary. It is unfortunate that the sample is so small and more teachers would have given a better evaluation of the design of the MOOC. Were the experts ex-teachers? did they have any pedagogical background or education? Have the 17 participants tried out the activities proposed in the MOOC? or did they just read them? Also, the resources were evaluated positively with some concerns about the quality of videos and subtitles. What about the proposed planning for activities? The 8 teachers should have given some feedback on that, specifically for the practical aspect of activities. It would be very useful to know this feedback.

6) Some of the information on findings is repeated in several sections. Try to synthesise as much as possible. There is no cross-referencing in the discussion of the results - were the results consistent with previous relevant research? 

7) Finally, what is the originality, the innovation of this paper? what does it contribute to knowledge? what are the implication for practice and education? These are all important points that should be addressed in the conclusions (last section - better to have the subheading 'Conclusions' instead of 'in summary'). 

References need some attention to follow APA7 rules in the text and in the end list. Check also p. 7 - first sentence (fully is used twice). Not sure why F in italics is used in the tables with the statistics - for sample we use the letter 'N'. 

Author Response

Thank you very much for carefully reading our paper and for your detailed comments and suggestions that helped us to complete and better clarify certain aspects. We believe that your contribution has helped us to improve our work.

Point 1: The first paragraph argues for the value of computational thinking/programming and robotics base on only one reference - more are needed.

Response 1: You are accurate in pointing out the lack of citations in the early part of the text. We have now included an extra 3 references from which we emphasize the importance of Bers' work (2008; 2023). This is important, not only because Bers is a renowned researcher in this area, but also because she has been involved in our current work and even acted as a consultant on the project this paper is based on.

The references added in this section are the following:

Mangina, E.; Psyrra, G.; Screpanti, L.; Scaradozzi, D. Robotics in the Context of Primary and Pre-School Education: A Scoping Review, IEEE Trans. Learn. Technol. 2023. doi: 10.1109/TLT.2023.3266631.

Bers, M. U.; Blake-West, J.; Kapoor, M. G.; Levinson, T.; Relkin, E.; Unahalekhaka, A.; Yang, Z. Coding as another language: Research-based curriculum for early childhood computer science. Early Childhood Res. Quart. 2023, 64, 394-404.

Bers, M. U. Blocks to robots: learning with technology in the early childhood classroom. Teachers College Press, New York, 2008. ISBN 978-0-8077-4847-3.

Point 2: The use of the word 'nursery' needs to be defined in terms of age of children as it is different in many countries. I would also recommend to try and use the word 'preschool' (as in the title of the project) or early childhood education. Also, some information on the educational system (EC and primary) in Portugal would be useful for helping the reader understand the context.

Response 2: The use of the term "nursery" was due to a translation error and has been replaced throughout the text by early childhood education or pre-school education. A brief reference was also made to the Portuguese education system, explaining the ages covered by preschool education and primary education (lines 142-146).

Point 3: Better to avoid the use of the word 'training' for teachers and replace it with professional development or professional learning to be consistent with relevant literature. 

Response 3:The term teacher training has been changed. We agree that it does not reflect the nature of the professional learning experience we are referring to. We have adopted the expression professional development, or professional learning programme as highlighted in the text.

Point 4: The authors mention a range of scholars and their work on designing a Mooc (e.g., Anderson, Moore on page 4 - and then Guardia et al. on page 6). I would suggest deleting the ones  that are not used in the presentation of the paper. Only Guardia et al. are sufficiently explained and related to the MOOC. Please also check the use of IET - TEI  on page 4.

Response 4: Both Anderson's constructs presented on page 4 and the guidelines proposed by Guardia et al. on page 6 were crucial for us to conceptualize the design of the MOOC. Nevertheless, we have revised the text (lines 171-173) to emphasize the importance of IET in online course design and included a pivotal reference (Miyazoe, T., & Anderson, T., 2013) that was inadvertently omitted from the original text. It is within this reference that the authors address the practical application of IET in the design of MOOCs:

Miyazoe, T., & Anderson, T. (2013). Interaction Equivalency in an OER, MOOCS and Informal Learning Era. Journal of Interactive Media in Education, 2013(2), Art. 9.DOI: https://doi.org/10.5334/2013-09

Point 5: There is no demographic info about the participants - only their gender. More details are necessary. It is unfortunate that the sample is so small and more teachers would have given a better evaluation of the design of the MOOC. Were the experts ex-teachers? did they have any pedagogical background or education? Have the 17 participants tried out the activities proposed in the MOOC? or did they just read them? Also, the resources were evaluated positively with some concerns about the quality of videos and subtitles. What about the proposed planning for activities? The 8 teachers should have given some feedback on that, specifically for the practical aspect of activities. It would be very useful to know this feedback.

Response 5: We have used a purposive sample, which is a non-probability sampling technique. Typically, a purposive sample, also known as judgmental or selective sampling, is a small sample where the researcher purposefully selects participants based on specific criteria relevant to the research objectives. The subjects are selected to the extent that the researchers believe they could be good informants in relation to the objectives of the study. So, the small purposive sample was selected based on three general profiles: early childhood educators, teachers, and specialists in the subject. For the first two profiles, no specific criteria, such as age, years of experience, or workplace location, were required. It was essential to include individuals from the broader universe of teachers and kindergarten educators working within the Portuguese education system. The experts who participated in the testing are domain specialists who work as teachers and researchers in higher education, either in Portugal or Spain, and are involved in implementing projects in the field of programming and robotics within their contexts.

Regarding the evaluation process, we have provided more detailed information in the Methodology section, aiming to facilitate a better understanding of the activities carried out by the testing participants. In any case, participants experienced the activities, albeit briefly, enabling us to make the necessary adjustments before launching the course.

Finally, the development team analysed the points highlighted by the participants to determine the best approach for implementing the required improvements. Childhood Educators and teachers provided positive feedback on the activities they tried, while the experts were more critical of certain activities, as reported in the results.

Point 6 and 7:

6) There is no cross-referencing in the discussion of the results - were the results consistent with previous relevant research? 

7) Finally, what is the originality, the innovation of this paper? what does it contribute to knowledge? what are the implication for practice and education? These are all important points that should be addressed in the conclusions (last section - better to have the subheading 'Conclusions' instead of 'in summary'). 

Responses 6 and 7: About questions 6 and 7, we have added a section of “Conclusions” where we have tried to answer those questions.

Point 8:  References need some attention to follow APA7 rules in the text and in the end list. Check also p. 7 - first sentence (fully is used twice). Not sure why F in italics is used in the tables with the statistics - for sample we use the letter 'N'. 

Response 8: For references and citations, one of the specific standards of the journal (American Chemical Society - ACS) was used, indicated in item 8.11 of the MPDI Style Guide (MDPI | Layout Style Guide). Regarding the term "fully", we have made the adjustment in the text, as indicated on page 7, the first item of the table.

The italic f is used as a standard in statistics for data frequency. The N is used as a standard in statistics for the size of the sample. We want to indicate the frequency of responses on the items considered.

Reviewer 2 Report

The presentation of the MOOK results is relatively straightforward and understandable. But in the present article, we do not see at all what were the educational activities in which the teachers participated. For example, table 1 and Week 2 in the Resources refer “Video and text with the "Principles of CT with/without the use of technology" “and “- Base texts and supplementary Reading- Artifacts with CT activities”. Ηere we would like a more detailed description. On week 3  (activities/Assessment) Basic text on ScratchJr programming - Planning of activities, and - Diverse activities for developing a project using the Scratch platform. Also here we would like a more detailed description. On week 3 Resources “other programming languages “  which? In week 4 where you talk about educational robotics, the sources only mention videos and texts. How can anyone teach robotics if they do not operate a robot themselves, even in a simulation? In general, I would like a little more detailed descriptions of the course content.

Author Response

Thank you very much for carefully reading our paper and for your comments and suggestions that helped us to complete and better clarify certain aspects. We believe that your contribution has helped us to improve our work.

Points: The presentation of the MOOK results is relatively straightforward and understandable. But in the present article, we do not see at all what were the educational activities in which the teachers participated. For example, table 1 and Week 2 in the Resources refer “Video and text with the "Principles of CT with/without the use of technology" “and “- Base texts and supplementary Reading- Artifacts with CT activities”. Ηere we would like a more detailed description. On week 3  (activities/Assessment) Basic text on ScratchJr programming - Planning of activities, and - Diverse activities for developing a project using the Scratch platform. Also here we would like a more detailed description. On week 3 Resources “other programming languages “  which? In week 4 where you talk about educational robotics, the sources only mention videos and texts. How can anyone teach robotics if they do not operate a robot themselves, even in a simulation? In general, I would like a little more detailed descriptions of the course content.

Responses: We have adjusted Table 1, providing additional details about the activities conducted in each Module.

Regarding the Educational Robotics Module, we recognized that it would pose a significant challenge due to the online and massive format of the course. Consequently, we chose to utilize unplugged robotics resources and online applications to ensure inclusivity for all participants. Additionally, we explored a variety of robots, showcasing their functionalities and practical applications. Given the nature of a massive course, it would be restrictive and exclusionary to require participants to possess specific resources. As a result, this module did not generate significant questions from the test participants.

Reviewer 3 Report

Dear author/s

The manuscript deals with a very interesting area. It is very clearly written and presents a pilot evaluation of a MOOC. Below I comment some concerns that I have about its elaboration.

The manuscript is self-referenced, as it is expected, as it deals with the specific MOOC evaluation. However, although the general structure of the MOOC is provided, the reader cannot fully realize details of the MOOC that are evaluated (e.g., content length and time ect.). So from this perspective, the scientific contribution of this evaluation is weak. It seems like a report that justifies some design restrictions and highlights next possible steps. A state of the art in the area might reveal relevant information at least in the discussion section. Based on the above, I propose a more detailed presentation of the MOOC characteristics so as to inform the reader about the merits of the pilot study and justify the contribution of the manuscript  to scholarship.    

Author Response

Thank you very much for carefully reading our paper and for your comments and suggestions that helped us to complete and better clarify certain aspects. We believe that your contribution has helped us to improve our work.

Points: The manuscript is self-referenced, as it is expected, as it deals with the specific MOOC evaluation. However, although the general structure of the MOOC is provided, the reader cannot fully realize details of the MOOC that are evaluated (e.g., content length and time ect.). So from this perspective, the scientific contribution of this evaluation is weak. It seems like a report that justifies some design restrictions and highlights next possible steps. A state of the art in the area might reveal relevant information at least in the discussion section. Based on the above, I propose a more detailed presentation of the MOOC characteristics so as to inform the reader about the merits of the pilot study and justify the contribution of the manuscript to scholarship. 

Responses:

- We have adjusted Table 1, in which we provide some details about the characterisation of the MOOC. In addition, in the discussion of the results, the tables provide details of the MOOC in each module that was evaluated by the test participants.

- About the scientific contribution of this evaluation, we have added a section of “Conclusions” where we have tried to answer those questions.

Reviewer 4 Report

The manuscript is well-organized with a clear logical analysis and robust statistical results. I have no revision suggestions to make, and I highly recommend publishing it in the current edition.

Author Response

Points: The manuscript is well-organized with a clear logical analysis and robust statistical results. I have no revision suggestions to make, and I highly recommend publishing it in the current edition.

Response: Thank you very much for reading our paper, for your comments and for your recommendation for publication. We have made some adjustments that have not changed the essence of the text and may help to clarify some aspects.